# Integrated economic and sexual and reproductive health programming among married and unmarried adolescent girls in Nigeria: Results from a quasi-experimental cohort study

Mary Phillips[1]*, Roselyn Odeh[2], Meghan Cutherell[1], Abednego Musau[3], Claire W. Rothschild[1], Albert Tele[4], Jenna Grzeslo[5], Alexis Coppola[1], Yusuf H. Wada[2], Noel Tonka[2], Alhaji Alhassan Bulama[2], Kehinde Atoloye[6], Olusesan A. Makinde[6]

1 Population Services International (PSI), Washington, DC, United States of America, 2 Society for Family Health (SFH) Nigeria, Abuja, Nigeria, 3 Population Services International (PSI), Nairobi, Kenya, 4 Independent consultant, Nairobi, Kenya, 5 BRAC International, Le Haye, The Netherlands, 6 Viable Knowledge Masters, Abuja, Nigeria

* mphillips@psi.org

## Abstract

Adolescent girls in sub-Saharan Africa are disadvantaged in nearly every measure of well-being when compared to their male peers, resulting in worse health outcomes and lower economic activity. Economically empowering adolescent girls is one approach to improving girls' wellbeing. Community-based groups that offer girls holistic support, such as increasing income generating capacity, fostering critical thinking, and building support for economic activity among their influencers, have been shown to improve economic and psychosocial assets. This paper describes the results of integrating economic empowerment programming into an existing sexual and reproductive health program for adolescent girls aged 15–19 in Kaduna and Ogun states in Nigeria. Outcomes of interest included increases in income generation and asset purchasing, improved decision-making around use of income and savings, and greater contribution to household expenditures as well as increases in contraceptive use and intention to use contraception in the future. The study was a quasi-experimental design consisting of an intervention group receiving the combined SRH and economic empowerment intervention and a concurrent comparison group receiving only the SRH intervention. Data was collected concurrently in both groups before participants were involved in the intervention (baseline) and nine months after (endline) for the same participants. Results varied by state, with exposure to the intervention associated with increases in the proportion of participants earning money (35pp increase in Kaduna, 58pp increase in Ogun) and almost equal increases in contributions to household expenses. Exposure to the intervention was associated with a significant increase in contraceptive use in Kaduna. Further, exposure to the intervention in Ogun was associated with significant increases in purchasing

which permits unrestricted use, distribution, and reproduction in any medium, provided the original author and source are credited.

**Data availability statement:** The data supporting the findings in this study are publicly available in Figshare under a CC BY 4.0 license under the title "A360 Economic Strengthening Pilot Evaluation Dat Sets" and DOI: https://doi.org/10.6084/m9.figshare.29390573.v1.

**Funding:** Individual authors did not receive specific awards. This work was supported by the Bill & Melinda Gates Foundation (https://www.gatesfoundation.org/) INV-004274 and the Children's Investment Fund Foundation (https://ciff.org/) R-1911-04245. Awards were given to Population Services International to support the overall program design, operations, and study. The funders had no role in study design, data collection and analysis, decision to publish, or preparation of the manuscript.

**Competing interests:** The authors have declared that no competing interests exist.

of assets and intent to use contraception in the future which were not observed in Kaduna. The findings suggest that layered economic empowerment programs can have positive outcomes for diverse cohorts of adolescent girls.

## Introduction

### Background

Adolescent girls in sub-Saharan Africa are disadvantaged in nearly every measure of well-being when compared to their male peers: girls are less likely to complete secondary school; more likely to be restricted in their movements; at higher risk for intimate partner violence; and experience lower perceptions of their own self-worth [1]. For instance, in Nigeria, more girls than boys drop out of junior and senior secondary school, 31% of girls are married prior to their 18th birthday while 37% have had a live birth before age 20 [2–4]. Higher rates of school dropout, and early marriage and childbearing occur in northern Nigeria than the national average. In Kaduna state, the median age at marriage is 16.6 years and age at first birth is under 19 years. In southern Nigeria, rates of child marriage and school dropouts are lower than in the north, but adolescent girls are at high risk for violence and face restrictions from negative gender and social norms. Consequently, more than 1 in 2 women in Ogun state have experienced physical violence [5]. School dropout and early marriage and childbearing contribute to higher fertility, worse health, and lower incomes [6]. Additionally, the children born to these women tend to experience low birth weight, lower rates of immunization, and higher risk of under-five mortality [7–9]. The consequences of harmful norms and practices, while dire for adolescent girls, are also detrimental to the communities and countries in which they live. Geographies where inequality is the greatest also experience the worst poverty and highest rates of conflict [10]. Gender inequalities limit women's participation in the labor force, reducing potential economic gains. Investments that keep girls in school and equip them for entry into the workforce in equal numbers to men could add USD 7 trillion to the global economy [11].

One approach for addressing gender inequity is to economically empower women and girls [12]. Economic empowerment can be understood as a process 'whereby women and girls experience transformation in power, agency, and economic advancement' [12]. Economic empowerment should encompass individual-level efforts that improve incomes and systematic changes made in partnership with women and girls to improve their capacity to exercise their economic power in ways that matter to them [13,14]. Economic empowerment interventions for girls can be categorized by those that offer direct economic training and support, such as vocational training, financial education, or employment programs; combined programs that layer in additional components such as life skills or sexual and reproductive health (SRH) information to compliment the financial component; and community-based girls' groups [15]. There is limited evidence on the effectiveness of vocational skills training specific to adolescents in low- and middle-income countries (LMICs). A

systematic review conducted in 2017 established small positive effects of vocational and business training on employment, including formal employment. However, the effects increased when programs included a gender component, and increases in earnings were greater if the program included life skills or an internship [16].

Life skills are a set of competencies derived from skills, knowledge, and attitudes that allow young people to thrive [17] by improving their ability to navigate psycho-social challenges and increasing resilience [18]. Valuable life skills include decision-making, negotiation, and communication skills [19–21]. Combining livelihood or vocational training with life skills is promising for increasing income, and contraceptive use [22]. Community-based girl groups (sometimes called safe spaces or girls' clubs) constitute girl-only environments enhanced with supportive mentors aiming to build girls' protective assets and improve outcomes in a range of domains [23]. Community-based girls' groups from LMICs have been established to result in positive effects on individual-level outcomes such as improvements in attitudes and beliefs about gender and health, improvements in economic and psychosocial assets, and increases in knowledge and awareness. However, outcomes that depend on external factors, such as behavior and health status, show mixed effects, with inconsistent improvements in contraceptive uptake or reductions in child marriage [24]. Complementary programming that works with girls' influencers to shift social norms is required to address the restrictions that preclude the attainment of intervention effects [24]. An example of such a program is the ENGINE program in Nigeria whose evaluation suggested that participation resulted in higher self-esteem, lower acceptance of violence, improved incomes, higher levels of engagement in income generating activities, and increased use of savings accounts, although results varied due to regional heterogeneity [25]. Despite the promising evidence, significant gaps exist in the understanding of what is effective, including the role of group dynamics on outcomes, the effect of savings on economic outcomes, differences in approaches for different cohorts of girls, and the optimal dosage of programming [24]. However, the current body of research suggests economic empowerment requires holistic investment in adolescent girls, with a focus on building their income generating capacity, as well as fostering critical thinking and building support for economic activity among their influencers.

In this paper we describe the results of integrating economic empowerment programming on top of an existing SRH program for adolescent girls aged 15–19 in two Nigerian states. We hypothesized that the combined intervention would lead to improvements in economic activity among girls, particularly increases in income generation and asset purchasing, improved decision-making around use of income and savings, and greater contribution to household expenditures as well as increase in the likelihood of contraceptive use and intention to use contraception in the future.

## Materials and methods

### Intervention background

This study was undertaken as part of the Adolescents 360 project, which aims to increase demand for, and voluntary use of modern contraception among adolescent girls aged 15–19 years [26]. The economic empowerment intervention was implemented between 2022–2023 in Kaduna in northern and Ogun in southern Nigeria. In each geography, girls in the intervention group participated in a four-session program designed to increase voluntary uptake of modern contraception and then a longer business skill development curriculum and vocational skills training. Participants in Kaduna were exclusively married girls while those in Ogun were primarily unmarried girls. Participants' first contact with the program was through the SRH intervention. In southern Nigeria this included classes for girls focused on menstrual hygiene, pregnancy, sexual health and rights, negotiation skills, decision-making and goal setting. In northern Nigeria, the curriculum also included personal hygiene and sexual health as well as improving family nutrition, interpersonal skills, and effective communication. Both interventions offered girls one-on-one sessions with health providers to discuss individual contraceptive preferences and provision of contraceptive services for those who chose to take up a method. The economic empowerment intervention included five business upskilling group sessions of approximately 90 minutes each where girls identified their strengths, set goals for the future, and received basic business training. Girls then chose up to two vocational skills,

 

one that was quick to master, such as soap-making, and one that required more time, such as tailoring or catering, to learn through an apprenticeship (Ogun) or through vocational training centers (Kaduna). Vocational training lasted four to five weeks depending on the skill selected. Girls opted to learn a variety of trades including catering, hair and make-up, poultry and fish farming, shoemaking and photography. During and after the vocational training, adolescent girls were provided with a mentor who offered support in developing and executing a business plan. The program culminated in a large, public graduation that doubled as a marketplace for adolescent girls to display products and services from their newly gained skills. Additional information on each program, including a comparison of what the standalone SRH intervention offered versus the layered economic empowerment intervention is described in Table 1.

## Research design

This study was conducted from June 2022 to April 2023. We employed a quasi-experimental design consisting of an intervention and a concurrent comparison group. For the intervention group, participants were adolescent girls aged 15–19 who received the combined SRH and economic empowerment intervention. A similarly sized cohort of adolescent girls of the same age group who only received an SRH intervention constituted the comparison group. Data was collected concurrently in both groups before participants were involved in the intervention (baseline) and nine months after (endline) for the same participants. Intervention clusters were the same locations where the intervention design was being implemented

**Table 1. Program elements by geography.**

| Geography | Northern Nigeria – SRH | Northern Nigeria – SRH+EE | Southern Nigeria – SRH | Southern Nigeria – SRH+EE |
|---|---|---|---|---|
| **Target Audience** | | | | |
| Sub-Geography | State: Kaduna | | State: Ogun | |
| Marital Status | Married | | Primarily unmarried | |
| Target Age | 15-19 | | 15-19 | |
| Location | Peri-urban/ Rural | | Urban | |
| School Status | Out-of-school/ In-school | | Out-of-school/ In-school | |
| **Program Dosage** | | | | |
| Group Size (number of participants per session) SRH Only; Economic Empowerment | 12 | 12; 24 | 12 | 12; 20 |
| Meeting Frequency | 2x weekly | 2x weekly | 2x weekly | 2x weekly |
| Session Duration | 90 minutes | 90 minutes | 90 minutes | 90 minutes |
| Program Duration | 2 weeks | 12 weeks | 2 weeks | 12 weeks |
| **Program Elements** | | | | |
| Life Skills | Yes | Yes | No | Yes |
| Mentorship/ Coaching | No | Yes | No | Yes |
| Social/ Girls' Group | Yes | Yes | No | Yes |
| Influencer outreach | Yes | Yes | Yes | Yes |
| Vocational Training | Yes[a] | Yes | No | Yes |
| Business Training | No | Yes | No | Yes |
| Financial Education | No | Yes | No | Yes |
| Savings | No | Yes | No | Yes |
| Linkage to Markets | No | Yes | No | Yes |
| SRH knowledge (contraception) | Yes | Yes | Yes | Yes |
| SRH access (contraception) | Yes | Yes | Yes | Yes |

[a] Light-touch approach of one 90-minute sessions on quick-to-learn skills such as beading and soap-making.

while comparison clusters were conveniently selected from the same state for easy accessibility by the research team but geographically distant from the intervention clusters to reduce spillover effects.

## Participants and sampling

**Recruitment.** Study participants were recruited by female program mobilizers promoting the SRH intervention using recruitment scripts. In Ogun, recruitment took place from May 24, 2022, to June 30, 2022. In Kaduna, recruitment took place from May 23, 2022, to June 30, 2022. Recruitment could take place in participants' homes, community or public spaces, or at the health facility. The female mobilizers originated from the communities where the SRH intervention was ongoing and were familiar with households where adolescents aged 15–19 years lived. At the initial contact with the adolescent girls, mobilisers requested to speak to them in private and administered the screening questions, which were built on the recruitment scripts. The inclusion criteria required that all participants be adolescent girls between the ages of 15–19, living in the selected study areas, willing to participate in the baseline and endline assessments, and with no plan to migrate out during the study period. Participants in northern Nigeria had to be married or living as if married to participate. An additional criteria for the intervention group was a willingness to participate in a three-month intervention, although specific details of the program content were not shared. During the recruitment, girls who were eligible but declined to participate in the study were still able to participate in the intervention activities. Eligible adolescent girls who expressed interest to participate in the study provided their phone contact details or physical addresses which were recorded in a confidential recruitment sheet and relayed to a trained female data collector. Female data collectors then contacted the individuals and scheduled an in-person meeting at the study site. Potential participants were informed by interviewers about the study objectives, time commitment, confidentiality, and any potential risks that could result from the study before they were asked to provide their consent to participate. Recruitment was continuous until the sample size was attained. The estimated sample size in each geography was 400 participants for each group which had been projected to be adequate to detect a modest average treatment effect of 10% difference-in-the-difference in the primary outcome variable (currently earning money) across the study groups between the two time points assuming 50% of participants in both groups were earning money at baseline at the 95% confidence level with 80% power. The sample was adjusted upwards to account for a 10% attrition rate for each group.

**Data collection.** The same structured questionnaire was used at baseline and endline. At endline, additional questions were included to document participants' exposure to economic empowerment programming. Trained female data collectors and supervisors were deployed to manage the field data collection. Data was collected using the Computer Assisted Personal Interview (CAPI) approach using the offline android application of the Open Data Kit (ODK) platform. All survey tools were translated from English into local languages (Hausa and Yoruba) and back-translated to ensure accuracy. Surveys were pre-tested with program participants in non-survey areas before data collection commenced to screen for survey programming errors and verify quality of data collectors. Scheduling for the endline surveys were conducted through phone calls or physical tracing through the female mobilisers who had recruited the participants. Interviews were conducted in-person except for Ogun where phone surveys were conducted in an attempt to reduce the attrition rates. Participants also received an incentive worth two US dollars in Nigeria naira after successfully completing the endline survey.

## Measures

The study examined outcomes in two domains – economic and SRH – which were measured at baseline and repeated at end-line using the same questions. Both domains were measured through self-reported responses in the survey questionnaires.

• **Economic domain measures:** The three measures for this domain were earning money (the study primary outcome), purchasing assets and contributing to household expenses. For earning money, all participants were asked '*Do you earn money*?', with possible close-coded responses of 'Yes', 'No', and 'Don't know.' For purchasing assets, all respondents were

given a list of nine assets, asked if they purchased each one and, if so, to respond with 'Yes' or 'No' to whether they used their own money to purchase the asset. Additionally, interviewers asked '*Have you used your own money to purchase an item that I did not list*?. To measure how frequently girls contributed to their household expenditure, participants were asked, '*In the last three months, how often did you contribute to household expenses*? Possible responses included, 'never, rarely, monthly, or weekly'. We created a dichotomous variable on contribution to household expenses, in which we coded weekly, monthly, and rarely responses as 1 (yes) while never, refused, and don't know responses were coded as 0 (no).

- **SRH domain measures:** The two measures for this domain were current contraceptive use and intent to use contraceptive in the future. To measure current contraceptive use, a slightly different approach was used for participants in Kaduna and Ogun. In Kaduna, since all participants were married, we assumed they had a potential need for modern contraception. To better understand the contraceptive needs of the participants in Ogun where girls were predominantly not married, we asked: '*Have you ever had sex*?'. All respondents in Kaduna and respondents from Ogun who had ever had sex were asked, '*Are you or your partner currently using any method to avoid pregnancy?*' To assess intention to use contraceptives in the future, current non-users of contraception irrespective of sexual activity were asked, '*Do you intend to use any method to avoid pregnancy in the next year?*'. Those who responded with a "No" were then asked, '*Do you intend to use any method to avoid pregnancy in the future?*'. A binary outcome referred to as intent to use contraception any time in the future was computed from the responses to these two questions for all current non-users.

In addition to the measures in the two domains, at endline, intervention group participants were asked six questions about their participation in the different intervention components. For components that took place over multiple sessions, responses were gathered on whether participants participated in none, some, or all sessions or replied with the specific number of sessions attended. For the comparison group, participants were asked, '*In the last nine months, have you participated in a program to help you reach your economic goals such as an apprenticeship program, vocational training, etc.?*' and the name of the program. For the specific questions asked, see S1 File.

## Ethical considerations

Approval for this study was received from the National Health Research Ethics Committee of Nigeria (Approval Number: NHREC/01/01/2007-06/05/2023B) and the Population Services International Research Ethics Board (09.2022). All participants 18 years and older and emancipated minors under 18 (married adolescents) provided written informed consent. Unmarried minors provided written informed assent in lieu of consent (southern Nigeria only).

## Data confidentiality and security

Password protected electronic devices were employed to capture survey responses using CAPI. Survey records were submitted to a secure institutional server daily or immediately if internet connection was available. Data was encrypted during transmission and the server was only accessible to authorized members of the research team.

## Inclusivity in global research

Additional information regarding the ethical, cultural, and scientific considerations specific to inclusivity in global research is included in the supporting information (S2 File).

## Statistical analyses

We used frequencies to describe participants' background characteristics and program dosage. Chi-square tests were conducted to identify variables where there were statistically significant differences between the intervention and comparison groups at baseline. Generalized Estimating Equation (GEE) models were fitted for the binary outcomes using the gaussian family distribution and identity link function, employing an exchangeable covariance structure using robust

standard errors. Linear models, also referred as linear probability models, can be used to assess absolute, rather than relative, changes in binary outcomes, providing valid inference with the use of robust standard errors even with known misspecification of the outcome distribution [27]. The GEE model included independent variables for study group (coded as a binary variable equal to 0 for the comparison group and 1 for the intervention group), time (coded as 0 for baseline and 1 for endline) and a group-time interaction term as the difference-in-differences (DiD) estimator. Model based estimates of the changes from baseline, standard errors (SE) and corresponding 95% confidence intervals (CIs) are provided along with *p*-values for documenting statistical significance. See S1 Table for the full model. We report results from the adjusted models that include covariates for baseline measures parameterized as follows: age (15–17 = 0; 18–19 = 1), education level (no formal school = 0; primary = 1; secondary = 3; above secondary = 3), marital status (not married = 0; married = 1) and parity (none = 0; 1 = 1 and 2 or more = 2) as categorical variables. The selection of the covariates was informed by literature on the factors that influence economic capabilities as well as variables that showed group differences at the baseline survey. Results of the analysis of the adjusted vs. unadjusted DiD are summarized in S2 Table.

## Results

### Sample characteristics

In Kaduna, 1,052 girls were mobilized to participate in the study (559 for the intervention group and 493 for the comparison group). Among these 1,049 (556 intervention, 492 comparison) were enrolled and surveyed at baseline, with three participants deemed ineligible in the intervention cohort due to being outside the target age range. At endline, 841 were interviewed (474 intervention, 367 comparison) resulting in follow-up rates of 85.3% and 74.4%, respectively. In Ogun, 927 girls were recruited (501 for the intervention group and 426 for the comparison group) and all enrolled and surveyed at baseline. At endline, 664 were interviewed (406 intervention, 258 comparison) resulting in follow-up rates of 81.0% and 60.6%, respectively. See Fig 1.

Table 2 shows the participant characteristics between intervention and control groups within geographies. In Kaduna, girls in the intervention group were significantly older than those in the comparison group, while in Ogun, girls in the intervention group were significantly younger. School-going rates and school achievement were high in both Nigerian geographies, although girls in the comparison group in Kaduna were significantly more likely to have no education than those in the intervention group. As designed, in Kaduna, all participants were married and in Ogun most participants were unmarried. Child-bearing rates followed the trend in marital status. Notable differences were observed within the geographies for parity, with the intervention group being significantly more likely to have given birth than the comparison group in Kaduna, but less likely in Ogun.

### Analysis of differential attrition

We conducted additional analysis to understand if there were significant differences on the primary outcomes at baseline between participants lost to follow-up (LTFU) and those who were not. In Kaduna, there were no significant differences across all outcomes. However, in Ogun there were significant differences between participants LTFU and those who were not in purchasing assets (p = .013) and earning money (p < .001) in the intervention group. Participants LTFU had higher rates of purchasing assets than those who were not (70.5% vs. 57.1%) and for earning money (42.1% vs. 19.5%). Baseline values for these two variables were incorporated as covariates in the adjusted DiD models, if they did not serve as an outcome variable. The full results of the differential analysis are presented in S3 Table.

### Program exposure

Overall, self-reported attendance across all key elements of the intervention was over 90% for participants in the intervention groups (see Table 3). A quarter of girls in the comparison group (28.3%) in Ogun reported that they had participated in any economic empowerment activity. Rates were higher in the Kaduna comparison group, where over half (59.1%)

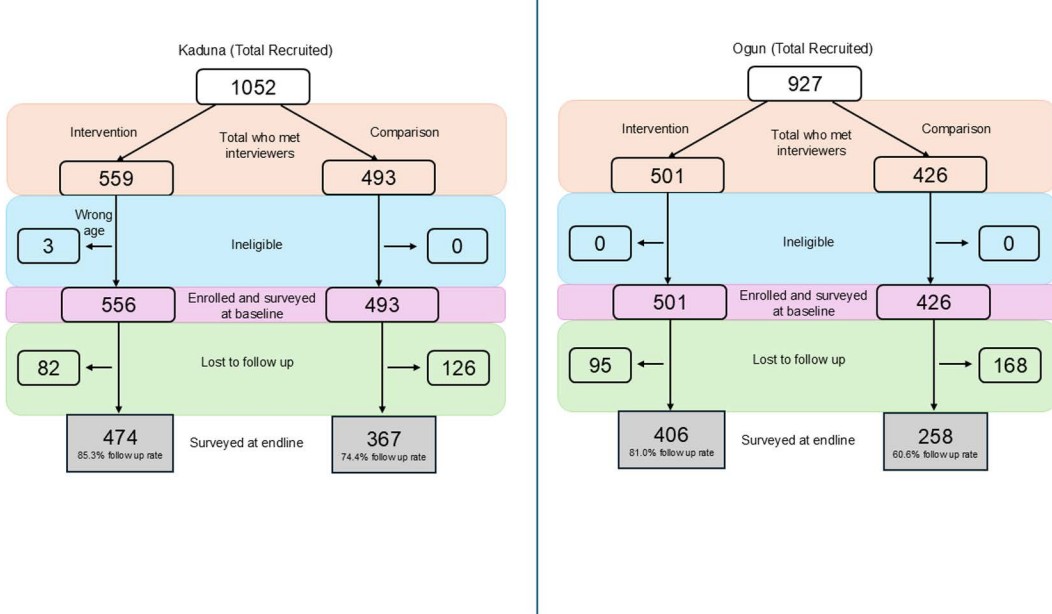

**Fig 1. Study flow diagrams for Kaduna and Ogun states.**

**Table 2. Participant socio-demographic characteristics for the intervention and comparison groups by geography.**

| Characteristic (at endline) | Ogun | | p-value | Kaduna | | p-value |
|---|---|---|---|---|---|---|
| | Intervention | Control | | Intervention | Control | |
| | n (%) | n (%) | | n (%) | n (%) | |
| **Age** | | | | | | |
| Age 15–17 | 222 (54.7%) | 112 (43.4%) | **0.005** | 29 (6.1%) | 59 (16.1%) | **<0.001** |
| Age 18–19 | 184 (45.3%) | 146 (56.6%) | | 445 (93.9%) | 308 (83.9%) | |
| **School Status** | | | | | | |
| Out-of-school | 248 (61.4%) | 159 (61.6%) | 0.950 | 128 (27.2%) | 119 (34.5%) | 0.026 |
| **Highest Level of Education** | | | | | | |
| No formal education | 1 (0.2%) | 0 (0.0%) | 0.838 | 11 (2.3%) | 91 (24.8%) | **<0.001** |
| Primary | 12 (3.0%) | 7 (2.7%) | | 9 (1.9%) | 57 (15.5%) | |
| Secondary | 367 (90.6%) | 237(91.9%) | | 368 (77.6%) | 193 (52.6%) | |
| Above Secondary | 25 (6.2%) | 14 (5.4%) | | 86 (18.1%) | 26 (7.1%) | |
| **Married/ Living as Married** | | | | | | |
| Yes | 17 (4.2%) | 21 (8.2%) | **0.032** | 474 (100.0%) | 367 (100.0%) | n/a |
| **Parity** | | | | | | |
| 0 | 308 (75.9%) | 159 (61.6%) | **<0.001** | 32 (6.8%) | 52 (14.2%) | **<0.001** |
| 1 | 31 (7.6%) | 32 (12.4%) | | 206 (43.5%) | 92 (25.1%) | |
| ≥2 | 67 (16.5%) | 67 (26.0%) | | 236 (49.8%) | 223 (60.8%) | |

reported participating in some sort of economic empowerment activity (see Table 3). In Kaduna, 174 (80.1%) of among the 217 girls participating in an apprenticeship program reported that the source was A360's SRH program, which contains low-dosage skills training.

## Economic outcomes

**Earns money.** In both geographies, the intervention group was significantly more likely to report earning money at endline than the comparison group as depicted in Figs 2 and 3. In Kaduna, exposure to the program was associated with a 35pp greater likelihood of self-reported earning money (95% CI [0.28; 0.43], p<0.001). Table 4 illustrates results for the three economic outcomes in both geographies. In Ogun, exposure to the program was associated with 58pp greater likelihood of self-reported earning money (95% CI [0.50; 0.67], p<0.001).

**Purchases assets.** In Kaduna, exposure to the intervention was not associated with a significant change in the proportion of adolescent girls who reported purchasing at least one asset with their own money in the preceding 12 months. Exposure to the program was associated with a non-significant 6pp increase in reporting the purchase of an asset (95% CI [−0.01; 0.13], p=0.103). In Ogun, exposure to the intervention was associated with a significant change in the proportion of participants reporting the purchase of an asset in the twelve months preceding the survey. Exposure was associated with a 31pp increase in reporting the purchase of an asset (95% CI [0.23; 0.38], p<0.001).

As a secondary outcome, we tracked the specific assets purchased to understand the types of items participants were spending their money on. In Kaduna at endline, a greater share of the intervention group had purchased livestock, kitchen utensils and home furniture than in the comparison group. At endline more participants in the comparison group reported the purchase of wrappers (hollandis) than those in the intervention group. Purchasing patterns were different in Ogun, where more participants in the intervention group reported purchases of phones, watches, and work tools at endline than in the comparison group. At endline more participants in the comparison group reported purchase of jewelry compared to the intervention group (results not shown).

**Contributes to household expenses.** In Kaduna, the intervention was associated with a statistically significant increase in contributing to household expenses probably due to decreases in contributions in the comparison group. Exposure to the program was associated with a 29pp greater likelihood of self-reported contribution to household expenses (95% CI [0.21; 0.36], p<0.001). Since this outcome included only participants who were earning money, it's worthwhile to consider the overall shifts in the total number of participants contributing to household expenses, as well as the proportions. Among the comparison group, 276 participants were contributing to household expenses at baseline and 271 were contributing at endline. Among the intervention group, 266 participants were contributing at baseline and 454 were contributing at endline. In Ogun, exposure to the intervention was associated with a statistically significant

**Table 3. Exposure to intervention activities among intervention group and any economic empowerment intervention among comparison group at endline.**

| Intervention Group | | Ogun | Kaduna |
|---|---|---|---|
| Participated in the life mapping/ goal setting | Yes | 398 (98.0%) | 466 (98.3%) |
| Participated in the primary package sessions (SRH) | All sessions | 350 (86.2%) | 431 (90.9%) |
| Participated in secondary package sessions (ES skills) | All sessions | 369 (90.9%) | 437 (92.2%) |
| Participated in vocational skills sessions | Yes | 397 (97.8%) | 463 (97.7%) |
| Number of one-on-one coaching sessions attended | None | 0 | 6 (1.3%) |
| | 1-2 | 287 (70.7%) | 133 (28.1%) |
| | >2 | 94 (23.2%) | 303 (63.9%) |
| | Refused | 25 (5.2%) | 32 (6.8%) |
| Number of group coaching sessions attended | 1-2 | 198 (48.8%) | 37 (7.8%) |
| | >2 | 187 (46.1%) | 434 (91.6%) |
| | Refused | 21 (5.2%) | 3 (0.6%) |
| **Comparison Group** | | | |
| Participated in an apprenticeship program, vocational training, etc., in the last nine months | Yes | 73 (28.3%) | 217 (59.1%) |

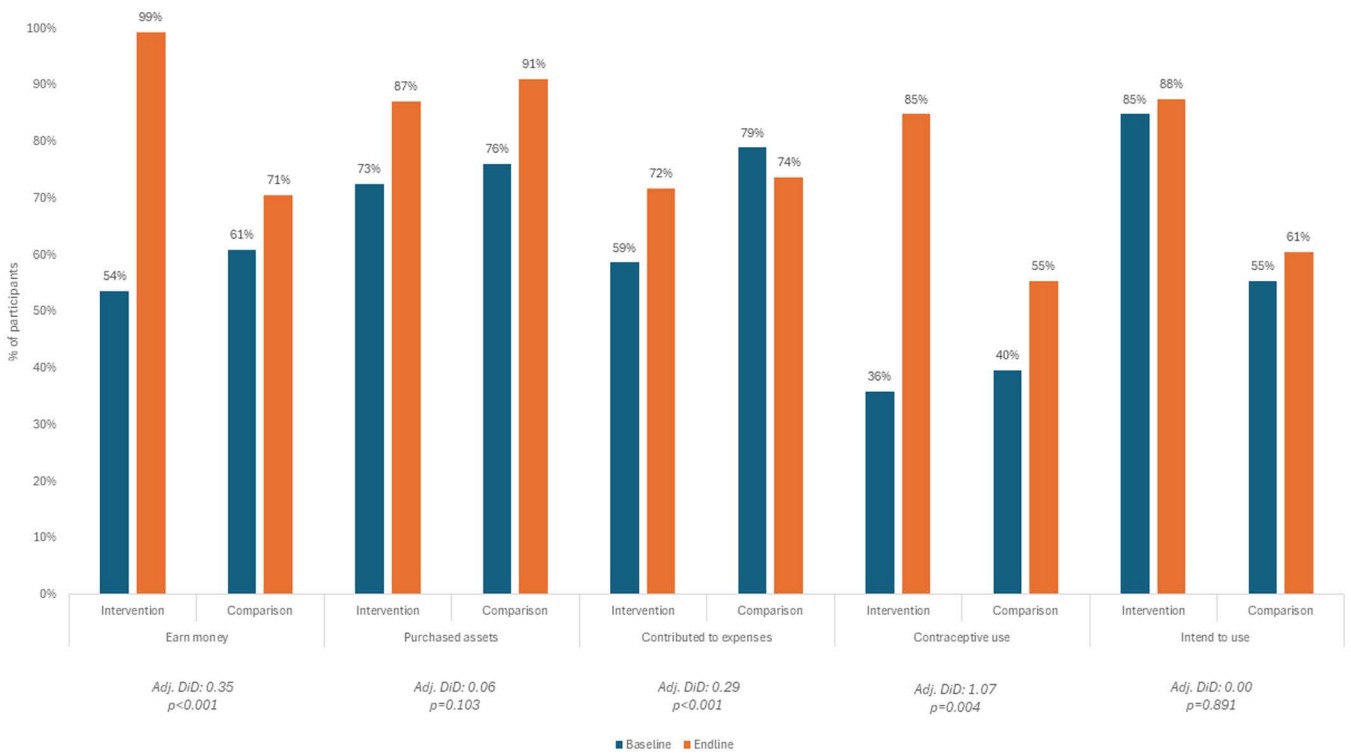

**Fig 2. Baseline and endline economic and SRH outcomes in Kaduna.**

increase in contribution to household expenses. There was a decline in the percentage of participants contributing to household expenses at endline, but the drop was relatively moderate in the intervention group. Exposure to the program was associated with a 28pp greater likelihood of contribution to household expenses among those earning money (95% CI [0.11; 0.45], p = 0.002). In Ogun, there was a notable increase in the proportion of overall participants contributing to household expenses at endline in the intervention group. At baseline 31 participants from the comparison group reported contributing to household expenses, compared to 33 at endline, 12.0% and 12.9% of all comparison group participants, respectively. Among the intervention group 29 participants reported contributing to household expenses at baseline and 119 reported contributing at endline, 7.1% and 29.4% of all intervention group participants, respectively.

### Reproductive health outcomes

**Sexual activity in Ogun.** In Ogun at baseline, about half of participants (49.8%) in the comparison group had ever had sex. At endline, this increased to 65.1%. Sexual activity rates were lower in the intervention group. Only 23.2% of participants reported ever having sex at baseline and this percentage increased only marginally at endline to 26.8%.

**Modern contraceptive use.** In Ogun at baseline, among participants who had ever had sex, 80.2% in the comparison group and 63.5% in the intervention group reported currently using a method of contraception. In both groups, current contraceptive use dropped at endline although the decline was less dramatic in the intervention group. At endline, current contraceptive use among participants who had ever had sex was 48.5% in the comparison group and 50.5% in the intervention group. Exposure to the program was associated with a non-significant 17.9pp (95% CI [−0.01; 0.37], p = 0.060) greater likelihood of current contraceptive use among participants who had ever had sex. In Kaduna at baseline 39.7% of participants in the comparison group and 35.9% of participants in the intervention group reported

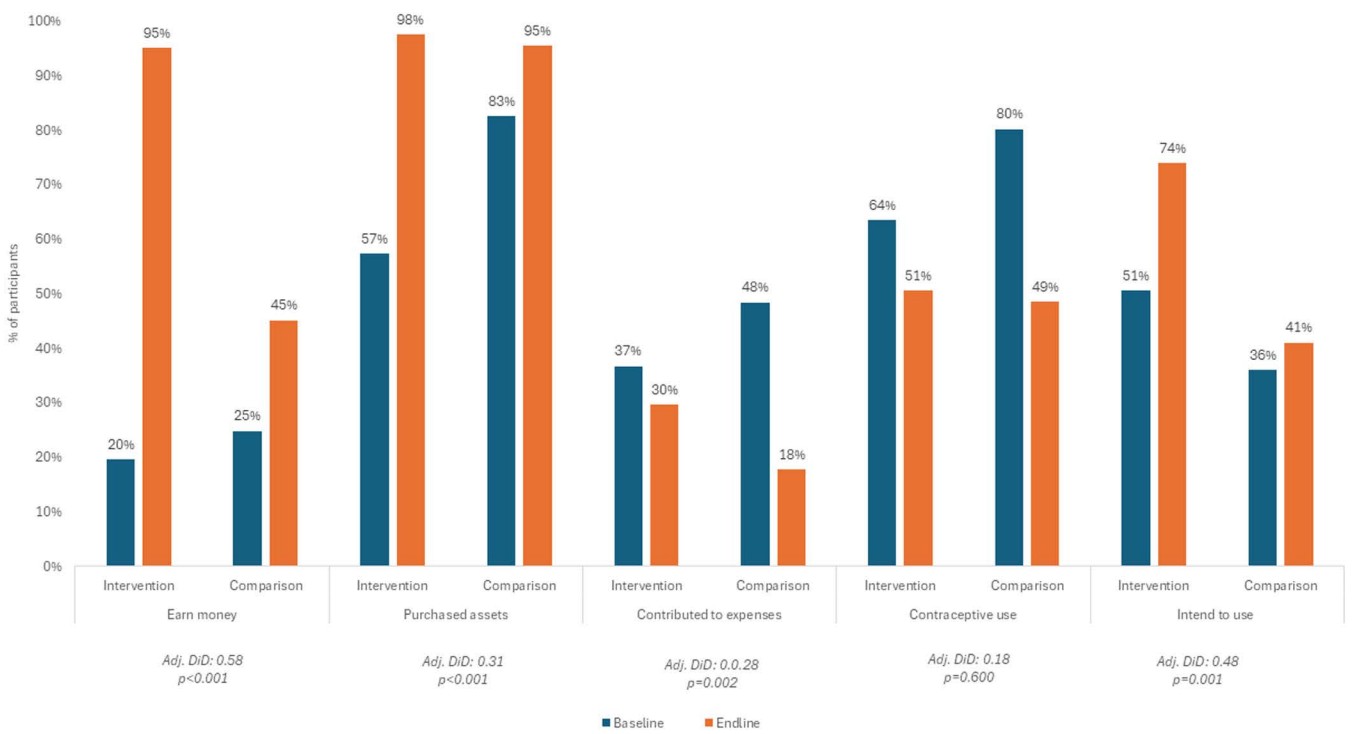

**Fig 3. Baseline and endline economic and SRH outcomes in Ogun.**

**Table 4. Adjusted difference-in-difference estimates for economic outcomes.**

| Kaduna | | | | Ogun | | | |
|---|---|---|---|---|---|---|---|
| | Baseline | Endline | Diff | | Baseline | Endline | Diff |
| **Earns Money** | | | | **Earns Money** | | | |
| Baseline: Intervention: (n = 556) Comparison (n = 493) | | | | Baseline: Intervention: (n = 406) Comparison (n = 258) | | | |
| Endline: Intervention: (n = 474) Comparison (n = 367) | | | | Endline: Intervention: (n = 405) Comparison (n = 255) | | | |
| Comparison | 60.85 | 70.57 | 9.72 | Comparison | 24.81 | 45.1 | 20.29 |
| Intervention | 53.6 | 99.37 | 45.77 | Intervention | 19.46 | 95.06 | 75.6 |
| *Adjusted DiD* | *0.35(0.28; 0.43)* | | ***p < 0.001*** | *Adjusted DiD* | *0.58 (0.50; 0.67)**** | | ***p < 0.001*** |
| **Purchases Assets** | | | | **Purchases Assets** | | | |
| Baseline: Intervention: (n = 554) Comparison (n = 493) | | | | Baseline: Intervention: (n = 405) Comparison (n = 258) | | | |
| Endline: Intervention: (n = 474) Comparison (n = 367) | | | | Endline: Intervention: (n = 406) Comparison (n = 258) | | | |
| Comparison | 76.06 | 87.19 | 11.13 | Comparison | 82.56 | 95.35 | 12.79 |
| Intervention | 72.56 | 91.14 | 18.58 | Intervention | 57.28 | 97.54 | 40.26 |
| *Adjusted DiD* | *0.06(−0.01; 0.13)* | | *p = 0.103* | *Adjusted DiD* | *0.31 (0.23; 0.38)**** | | ***p < 0.001*** |
| **Contributes to household expenses** | | | | **Contributes to household expenses** | | | |
| Baseline: Intervention: (n = 298) Comparison (n = 300) | | | | Baseline: Intervention: (n = 79) Comparison (n = 64) | | | |
| Endline: Intervention: (n = 472) Comparison (n = 297) | | | | Endline: Intervention: (n = 402) Comparison (n = 187) | | | |
| Comparison | 79 | 73.74 | −5.26 | Comparison | 48.44 | 17.65 | −30.79 |
| Intervention | 58.72 | 71.82 | 13.1 | Intervention | 36.71 | 29.6 | −7.11 |
| *Adjusted DiD* | *0.29(0.21; 0.36)**** | | ***p < 0.001*** | *Adjusted DiD* | *0.28 (0.11; 0.45)*** | | ***p = 0.002*** |

currently using contraception. At endline, 55.4% of participants in the comparison group and 84.9% of participants in the intervention group reported current contraceptive use. Exposure to the program was associated with a 207pp increase in contraception use (95% CI [0.67; 3.48], p = 0.004).

**Intent to use modern contraception.** In Kaduna at baseline 55.4% and 84.9% of participants not currently using a contraceptive method in the comparison and intervention groups, respectively, reported an intent to use contraception in the future. At endline, 60.5% and 87.5% of participants not currently using a method in the comparison groups and intervention groups, respectively, reported the same. Exposure to the program was not associated with an increase in intent to use contraception. In Ogun at baseline, 36.0% of participants in the comparison group and 50.5% of participants in the intervention group not currently using contraception reported an intent to use contraception in the future. At endline, these figures increased so that 41.0% of current non-users in the comparison group and 73.9% of current non-users in the intervention group were intending to use contraception in the future. Exposure to the program was associated with a 48pp increase in intent to use contraception in the future among current non-users (95% CI [0.21; 0.75], p = 0.001).

For all reproductive health outcomes, see Table 5.

## Discussion

The findings suggest that the layered economic empowerment intervention led to statistically significant, positive outcomes for adolescent girls in both geographies. Results were especially pronounced for the outcome on earning money in both Kaduna and Ogun. Our findings align with previous assessments on the potential to improve earnings through interventions that promote financial education or vocational and/or business skills interventions [22,28]. While earning is an important first step on the pathway to economic empowerment, it is necessary to understand how this money is being used and resulting changes in behavior or power at the household and community level. In Kaduna, exposure to the intervention had no effect on purchasing assets, while in Ogun exposure led to a 31pp increase in purchasing assets. Although there was no significant effect in Kaduna, over 90% of girls in the intervention group had purchased an asset in the 12 months preceding endline. The intervention effect detected in Ogun was attributable to a larger change in the intervention group of the proportion of girls who had purchased an asset between baseline and endline. Notably, the intervention group in Ogun constituted relatively younger girls who might not have had opportunities to purchase assets prior to the intervention because of weak economic capabilities. Interestingly, program participation was associated with higher likelihood of buying assets that support income generating activities – such as livestock in Kaduna and "work tools" in Ogun. These

**Table 5. Adjusted difference-in-difference estimates for SRH outcomes.**

| Kaduna | | | | Ogun | | | |
|---|---|---|---|---|---|---|---|
| | Baseline | Endline | Diff | | Baseline | Endline | Diff |
| **Contraceptive use** | | | | **Contraceptive use** | | | |
| Baseline: Intervention: (n = 543); Comparison (n = 484) | | | | Baseline: Intervention: (n = 74); Comparison (n = 121) | | | |
| Endline: Intervention: (n = 474); Comparison (n = 366) | | | | Endline: Intervention: (n = 109); Comparison (n = 167) | | | |
| Comparison | 39.67 | 52.73 | 13.06 | Comparison | 80.17 | 48.5 | −31.67 |
| Intervention | 35.91 | 89.87 | 53.96 | Intervention | 63.51 | 50.46 | −13.05 |
| *Adjusted DiD* | *2.07(0.67; 3.48)* | | *p = 0.004* | *Adjusted DiD* | *0.18 (−0.01; 0.37)* | | *p = 0.600* |
| **Intends to use contraception** | | | | **Intends to use contraception** | | | |
| Baseline: Intervention: (n = 344); Comparison (n = 280) | | | | Baseline: Intervention: (n = 356); Comparison (n = 111) | | | |
| Endline: Intervention: (n = 48); Comparison (n = 172) | | | | Endline: Intervention: (n = 351); Comparison (n = 173) | | | |
| Comparison | 55.36 | 60.47 | 5.11 | Comparison | 36.04 | 41.04 | 5 |
| Intervention | 84.88 | 87.5 | 2.62 | Intervention | 23.88 | 72.93 | 49.05 |
| *Adjusted DiD* | *0.00(−0.14; 0.14)* | | *p = 0.891* | *Adjusted DiD* | *0.48 (0.21; 0.75)* | | *p = 0.001* |

findings suggest a cycle of economic empowerment, in which early earnings are successfully invested in growing the income generating activity. This use of income is encouraging, given previous research that suggests women often face more pressure to spend money on urgent family needs, leaving less for investments in business growth [29]. The authors speculate that the work done on goal setting and sharing goals with key influencers (husbands and parents) may have given girls the tools (persuasion, planning, and confidence) required to negotiate use of their earnings.

In both geographies, exposure to the intervention was associated with increased contributions to household expenses. For girls and young women, contributing to household expenditure may be an important way to gain status and reduce instances of interpersonal violence [30]. The proportion of girls contributing to household expenses in the previous three months in both the intervention and comparison groups was much higher in Kaduna than in Ogun, which probably reflects their status as married women who have a responsibility to contribute to overall household well-being versus unmarried girls who may face fewer obligations to support the household. In Ogun contributions dropped from baseline to endline in both the comparison and intervention group, but the magnitude of the change was much smaller in the intervention group, resulting in a positive estimate of program impact. This may suggest that the intervention offered some protective properties that enabled girls to keep contributing to their household, despite external factors that made it harder to do so. For instance, economic contributions among unmarried girls may have been affected by the restart of the school year and limited opportunities to engage in income generation.

Findings on SRH outcomes are mixed, which aligns with findings from other multi-component programs [31]. In this study, all participants (intervention and comparison) received the SRH intervention and so the study was designed to measure the incremental effect of layering on the economic empowerment component. The clearest benefits were in Kaduna, where exposure to the intervention was associated with a 200pp increase in self-reported contraception use, and nearly 90% of intervention group respondents reported using a method of contraception at endline. Kaduna state, in general, has low rates of contraceptive use among married adolescents and high birth rates [5]. These results show the potential of combined economic and health programming that helps girls understand the connection between health seeking behaviors (in this case contraceptive uptake) in pursuit of larger life goals, such as income generation. In Ogun, exposure to the program was not associated with increases in contraceptive use among sexually active girls. For many girls in the south, contraceptive use may not be a relevant practice: only 1 in 4 girls in the intervention group and 1 in 2 girls in the comparison group reported they had ever had sex at endline. Given that we did not capture information on frequency and recency of sexual activity, our measurement approach may overestimate the proportion of girls who are at risk of pregnancy. As has been discussed extensively elsewhere, contraceptive uptake among unmarried girls, who often have infrequent, covert sex is a challenge for public health programming [32]. Furthermore, evidence suggests that younger and unmarried girls are less likely to benefit from the SRH components for combined interventions because of a limited understanding of sexuality and reproduction but also because session facilitators might face personal discomfort and technical difficulties covering SRH content [28]. There is a need to support girls who have intermittent sex and therefore inconsistent need for contraception in a way that is in line with their preferences and to develop metrics that better respond to patterns of intermittent need. In Kaduna, there was no associated effect of program exposure on intent to use contraception, as intent to use stayed relatively high (over 85%) among the intervention group between baseline and endline. In Ogun, however, exposure to the program resulted in a 48pp increase in reported intent to use contraception. The integrated intervention provided girls additional contact with mentors to learn more about SRH, to discuss their concerns and to receive positive reinforcement about the role of contraception as a tool for pursuing their self-defined goals, motivating their willingness to use contraception in the future. Contrastingly, for the girls from Kaduna who were married contraception was relevant and those who desired to use contraception might already have adopted a method leaving only those pursuing motherhood to respond to this question. While intent does not guarantee future contraceptive use, it is a promising signal that girls in the combined program model see the relevance of contraception for attaining their goals and may be primed to choose to take up a method in the future when they do determine that they are in need.

                                                                13 / 17

There are several limitations that should be considered when interpreting the results of this study. First, because allocation of intervention sites had already been determined at the time of study design, we used a quasi-experimental, non-randomized evaluation design. Further, the use of program mobilizers for participant recruitment could have introduced a selection bias specifically for adolescent girls joining the intervention group. For instance, mobilisers could have preferred to recruit relatives, daughters of their friends or girls of specific religious, ethnic or clan affiliations. To mitigate the effects of this bias, we examined group differences using baseline data and employed any demographic variables where there was a difference as covariates in our DiD analysis. While our DiD approach accounts for time-invariant confounding (both measured and unmeasured), there is a possibility of unmeasured confounding by time-varying factors differentially impacting the intervention and comparison groups. In our study there were only nine months between the baseline and endline studies, which is a short period of time to show the sustainability of the observed changes in economic empowerment and SRH outcomes. A longer time period may have allowed us to show stronger intervention effects or provided more confidence about the long-term durability of the effects seen initially. The attrition at endline, especially in Ogun state, may have biased our final results towards girls who had more need for the program and were therefore faster to show initial results. We addressed this limitation by including baseline values of the factors that showed significant differences between the intervention and comparison groups and those retained and LTFU in our analysis.

## Conclusion

Our findings provide relevant insights to guide economic empowerment programming that addresses the needs of different girls in diverse settings. While the study populations were drawn from the same age cohort and country they represent important differences within sub-populations of adolescent girls. A key contribution of this study is its inclusion of diverse sub-groups of adolescent girls – a population that is often, inaccurately, considered to be homogenous across sub-Saharan Africa. The study assessed program effectiveness among married and unmarried girls, differing levels of education attainment, urban and rural, with children and without, and in different cultural contexts. What is notable is the consistency of positive impacts in the key dimensions measured, despite these differences. As discussed above, given their differences in marital status, the social role placed on the girls in Kaduna and the girls in Ogun is likely quite different. In Kaduna, married girls may be viewed socially as women, with responsibility to care for their children and home. They are also greatly constrained by gender norms and lack the freedom to leave the home or seek work without permission from their husbands. The program addressed this barrier by engaging with community leaders who showed support of the girls' economic activities and provided an enabling environment for their participation in income generation. Given their status as adults with responsibility, once there was general social approval, girls were highly motivated to contribute to their own households. In Ogun, by contrast, unmarried girls are likely still viewed as children, with limited capacity to contribute economically and under the restrictions of their parents. Again, by working with key influencers in the community the program was able to generate support for these girls and help shift perspectives about their social role. Many girls in Ogun chose to contribute their money earned to school expenses, a goal supported by unmarried girls' influencers. Understanding the specific barriers in place for girls in each context and incorporating thoughtful program elements to address those barriers can mean the difference between success and failure.

## Supporting information

**S1 File. Program exposure questions.**
(DOCX)

**S1 Table. GEE model.**
(XLSX)

**S2 Table. Comparison of unadjusted and adjusted DiD.**
(DOCX)

**S3 Table. Differential attrition analysis table.**
(DOCX)

**S2 File. Questionnaire on inclusivity in global research.**
(DOCX)

## Acknowledgments

The authors are grateful to the staff of the Royal Heritage Health Foundation (RHHF) and Society for Women Development and Empowerment in Nigeria (SWODEN) who were implementation partners for this program. We acknowledge the contributions of the members of the Society for Family Health Nigeria who oversaw delivery of this program and the contributions of the Nigerian-based research team Viable Knowledge Masters. We acknowledge the adolescent girls who took the time to participate in the survey and appreciate their willingness to contribute to this work.

## Author contributions

**Conceptualization:** Mary Phillips, Meghan Cutherell, Abednego Musau, Alexis Coppola.

**Data curation:** Jenna Grzeslo, Olusesan A. Makinde.

**Formal analysis:** Abednego Musau, Claire W. Rothschild, Albert Tele, Jenna Grzeslo, Kehinde Atoloye, Olusesan A. Makinde.

**Funding acquisition:** Mary Phillips, Meghan Cutherell, Alexis Coppola.

**Investigation:** Olusesan A. Makinde.

**Methodology:** Mary Phillips, Meghan Cutherell, Abednego Musau, Claire W. Rothschild, Jenna Grzeslo, Olusesan A. Makinde.

**Project administration:** Roselyn Odeh, Meghan Cutherell, Abednego Musau, Jenna Grzeslo, Yusuf H. Wada, Noel Tonka, Alhaji Alhassan Bulama, Kehinde Atoloye, Olusesan A. Makinde.

**Software:** Albert Tele.

**Supervision:** Roselyn Odeh, Meghan Cutherell, Abednego Musau, Jenna Grzeslo, Yusuf H. Wada, Noel Tonka, Alhaji Alhassan Bulama.

**Writing – original draft:** Mary Phillips, Abednego Musau.

**Writing – review & editing:** Mary Phillips, Roselyn Odeh, Meghan Cutherell, Abednego Musau, Claire W. Rothschild, Jenna Grzeslo, Alexis Coppola, Yusuf H. Wada, Noel Tonka, Alhaji Alhassan Bulama, Kehinde Atoloye, Olusesan A. Makinde.

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
