## [Decision Letter · Decision Letter 0]

3 Dec 2024

We look forward to receiving your revised manuscript.

Kind regards,

Ibrahim Jahun, MD, MSC, PhD

Academic Editor

PLOS ONE

Journal Requirements: 

2. Please include a complete copy of PLOS’ questionnaire on inclusivity in global research in your revised manuscript. Our policy for research in this area aims to improve transparency in the reporting of research performed outside of researchers’ own country or community. The policy applies to researchers who have travelled to a different country to conduct research, research with Indigenous populations or their lands, and research on cultural artefacts. The questionnaire can also be requested at the journal’s discretion for any other submissions, even if these conditions are not met.  

Please find more information on the policy and a link to download a blank copy of the questionnaire here: https://journals.plos.org/plosone/s/best-practices-in-research-reporting. 

Please upload a completed version of your questionnaire as Supporting Information when you resubmit your manuscript.

3. Please note that funding information should not appear in the Acknowledgments section or other areas of your manuscript. We will only publish funding information present in the Funding Statement section of the online submission form. Please remove any funding-related text from the manuscript. 

4. Please include a separate caption for figure 1 in your manuscript.

5. We are unable to open your Supporting Information file [S5 Dataset.dta, S6 Dataset.dta]. Please kindly revise as necessary and re-upload.

**Additional Editor Comments:**

Financial disclosure:

Please add “findings are discussion are the opinions of the authors and do not reflect the position of the sponsors”.

Introduction:

This section is too long. Even though no word limit provided in author’s guide, there is need to be concise and direct to the points relating to the objectives of the paper. The section now spans to about 5 pages and will be better if the section is concisely reduced to 2.5 pages. This will enable readers not to be lost regarding the study objectives.The section will also benefit from basic statistics about reproductive health indicators among young girls globally and in Nigeria as well as statistics about poverty in this population (when compared to young men counterparts). The section should read like a scientific paper with facts (data) and not like a book.

Methods:

Recruitment: It is confusing how 400 participants relate to Table 1, whereby 12 groups and 24 upskills were reported. Please make this clearer or reconcile.Sample size: it is unclear how the sample size of 400 was reached. Please elaborate.Measures: it will be good to clearly list and define all variables that were collected. Please group these variables in to baseline and endline to enable proper comprehension.Data confidentiality and security: this subsection is missing. Please provide the subsection.

Results:

Failure to clearly define variables of interests in the methods makes it difficult to follow and understand the results. There is abundant data and information that should be strategically presented. Consider reducing the length of the results section by referring the readers to appropriate tables for details. Lengthy results make it difficult to follow and link it to discussion and conclusion sections. Also consider using appropriate visuals to show comparison of key findings to enable comparison. This may also reduce the length of the results.

Discussion and conclusion:

Please split the section into “Discussion” and “Conclusion and Recommendation”.Will be good to be concise and to ensure discussion is within the scope and findings of this study.

**Reviewers' comments:**

Reviewer's Responses to Questions

**Comments to the Author**

1. Is the manuscript technically sound, and do the data support the conclusions?

Reviewer #1: Yes

Reviewer #2: Partly

2. Has the statistical analysis been performed appropriately and rigorously?

Reviewer #1: Yes

Reviewer #2: Yes

3. Have the authors made all data underlying the findings in their manuscript fully available?

Reviewer #1: Yes

Reviewer #2: Yes

4. Is the manuscript presented in an intelligible fashion and written in standard English?

Reviewer #1: Yes

Reviewer #2: Yes

Reviewer #1: The study addresses an important gap by examining the effects of integrated economic empowerment and sexual and reproductive health (SRH) interventions on adolescent girls in Nigeria. This is particularly valuable given the unique challenges faced by this demographic.

However there are some improvements I would recommend:

1. Abstract Clarity: The abstract is dense, especially in terms of results. It would be beneficial to streamline the findings to highlight the most impactful outcomes.

2. Introduction - Background Context: The background is comprehensive but could benefit from a more focused framing of the economic and SRH challenges specifically faced by adolescent girls in Nigeria. Additionally, contextualizing this intervention in the broader scope of Sub-Saharan Africa would help readers unfamiliar with the region.

3. Methodological Details:

• Sampling and Recruitment: More details about the sampling process, including any potential biases, would enhance the transparency and replicability of the study.

• Covariates and Confounding: The manuscript mentions adjustments for certain covariates. It would be helpful to clarify why specific variables (e.g., age, education) were chosen and to justify their relevance in the analysis.

4. Discussion and Implications:

• Interpretation of Findings: The discussion could benefit from more critical reflection on why certain outcomes varied significantly between Kaduna and Ogun, particularly regarding SRH outcomes. Exploring cultural, social, or structural factors could provide readers with a more nuanced understanding.

• Program Duration and Sustainability: The study was conducted over a nine-month period, which limits long-term outcome assessment. Discussing the sustainability of these outcomes beyond the study period would be valuable.

5. Typographical Errors: A thorough proofreading would help eliminate minor typographical errors and improve readability.

Reviewer #2: 1. Expand the Timeline

• The study observed changes over nine months, which might not fully capture the long-term effects of the intervention. Extending the study period would provide more robust insights into sustained outcomes, particularly in economic empowerment and contraceptive use.

2. Address Attrition Issues

• Attrition rates were high, especially in Ogun. Future studies could employ strategies such as incentivizing participation, increasing community engagement, or using mobile follow-ups to reduce loss to follow-up and ensure a more representative sample at endline.

3. Enhance Measurement of SRH Outcomes

• Include detailed questions on sexual activity (e.g., frequency and recency) to better estimate contraceptive needs among participants. Additionally, qualitative methods like focus group discussions could uncover deeper reasons for changes in SRH behaviors.

4. Explore Regional Variability

• The study noted significant differences in outcomes between Kaduna and Ogun. Future research should delve into the contextual factors, such as cultural norms and economic conditions, to tailor interventions more effectively for each region.

5. Incorporate Broader Indicators of Empowerment

• While economic empowerment was assessed through earnings, asset purchases, and contributions to household expenses, consider adding measures like financial independence, agency in decision-making, and changes in social norms as broader indicators.

6. Strengthen Comparisons

• In non-randomized designs, controlling for potential confounding variables is critical. Future studies could implement propensity score matching or more sophisticated econometric techniques to reduce bias in comparisons between intervention and control groups.

7. Consider Mixed Methods

• Quantitative surveys provide key outcome measures, but qualitative data can enrich the understanding of barriers, motivators, and participants' perceptions. Adding interviews or focus groups could offer deeper insights into program impacts.

8. Diversify Economic Empowerment Activities

• Tailor vocational training to include more high-demand, scalable skills, and expand mentorship duration to ensure participants gain practical, sustainable income-generating capabilities.

9. Engage Influencers More Effectively

• The study showed promising results from engaging community leaders. Future interventions could expand this engagement to include more extensive workshops for parents, husbands, and community gatekeepers to amplify support for girls' empowerment.

10. Consider Scalability

• Evaluate cost-effectiveness and scalability of the combined intervention, ensuring that successful elements can be adapted and expanded to other regions or country

**Do you want your identity to be public for this peer review?** For information about this choice, including consent withdrawal, please see our Privacy Policy

Reviewer #1: **Yes: ** AMOS M'YISA MAKELELE

Reviewer #2: No

---

## [Author Response · Author response to Decision Letter 1]

29 Jan 2025

Editorial Review

The revised manuscript has been formatted aligning with the PLOS ONE style guidance

2. Please include a complete copy of PLOS’ questionnaire on inclusivity in global research in your revised manuscript. Our policy for research in this area aims to improve transparency in the reporting of research performed outside of researchers’ own country or community. The policy applies to researchers who have travelled to a different country to conduct research, research with Indigenous populations or their lands, and research on cultural artefacts. The questionnaire can also be requested at the journal’s discretion for any other submissions, even if these conditions are not met.  

Please find more information on the policy and a link to download a blank copy of the questionnaire here: https://journals.plos.org/plosone/s/best-practices-in-research-reporting. 

Please upload a completed version of your questionnaire as Supporting Information when you resubmit your manuscript.

The completed inclusivity in global research questionnaire has been submitted as a supporting information file (S7 File)

3. Please note that funding information should not appear in the Acknowledgments section or other areas of your manuscript. We will only publish funding information present in the Funding Statement section of the online submission form. Please remove any funding-related text from the manuscript. 

This section has been deleted from the revised manuscript

4. Please include a separate caption for figure 1 in your manuscript.

The caption for Fig 1 has been included in the revised manuscript

5. We are unable to open your Supporting Information file [S5 Dataset.dta, S6 Dataset.dta]. Please kindly revise as necessary and re-upload.

We thank the editorial team for catching this oversight. We have resubmitted the data files as CSV files (S5 File and S6 File) which should be able to open with the Microsoft suite

Additional Editor Comments:

Financial disclosure:

Please add “findings are discussion are the opinions of the authors and do not reflect the position of the sponsors”.

This has been updated in the submission platform as recommended

Introduction:

This section is too long. Even though no word limit provided in author’s guide, there is need to be concise and direct to the points relating to the objectives of the paper. The section now spans to about 5 pages and will be better if the section is concisely reduced to 2.5 pages. This will enable readers not to be lost regarding the study objectives.

Thank you for this suggestion. We have reduced the introduction section to fit into two and half pages.

The section will also benefit from basic statistics about reproductive health indicators among young girls globally and in Nigeria as well as statistics about poverty in this population (when compared to young men counterparts). The section should read like a scientific paper with facts (data) and not like a book.

We have enriched the introduction with more statistics citing the most recent Multiple Integrated Cluster Surveys of 2021 and the Nigeria Demographic Health Survey 2023/2024 on page 3.

Methods:

Recruitment: It is confusing how 400 participants relate to Table 1, whereby 12 groups and 24 upskills were reported. Please make this clearer or reconcile.

We thank the reviewer for the need to clarify this point. The SRH sessions for both the comparison (SRH only) group and the intervention (SRH and Economic Empowerment) group in both states consisted of a group of 12 adolescent girls. The economic empowerment sessions were nearly twice large in Ogun (sessions of 20 participants each) or equal to two times as large in Kaduna (sessions of 24 participants each). This clarification has been provided in Table 1 on page 8.

Sample size: it is unclear how the sample size of 400 was reached. Please elaborate.

We thank the editor for raising this important point. We estimated the sample size to detect a 10% difference in difference in the primary outcome variable (currently earning money) between the intervention and comparison groups and between the two study time points at the 95% confidence level assuming 50% in both groups were earning money at baseline with 80% power and a 10% attrition rate. This statement has been added in the sub-section on participant recruitment and sampling on page 10.

Measures: it will be good to clearly list and define all variables that were collected. Please group these variables into baseline and endline to enable proper comprehension.

We regret the lack of clarity on the study measures. Both baseline and endline included the same measures. Broadly, the measures were categorized into two domains (a) economic and (b) sexual and reproductive health. There were three measures for the economic domain, and all were binary variables - earning money, purchasing an asset using one's own money and contributing to household expenses. There were two measures for the SRH domain - current use of contraception and intention to use contraception in the future which were also binary variables. We have made revisions on page 11 and 12 to respond to this feedback.

Data confidentiality and security: this subsection is missing. Please provide the subsection.

We have included statements on this on page 13 starting on line 242.

Results:

Failure to clearly define variables of interests in the methods makes it difficult to follow and understand the results. There is abundant data and information that should be strategically presented. Consider reducing the length of the results section by referring the readers to appropriate tables for details. Lengthy results make it difficult to follow and link it to discussion and conclusion sections. Also consider using appropriate visuals to show comparison of key findings to enable comparison. This may also reduce the length of the results.

We regret the confusion the lack of clarity on the study measures affected the ease of following the results. We have addressed this concern in two ways. First, we have specified the measures in the methods section. Second, we have reduced the amount of text in the results section, allowing the reader to engage with the bulk of the results in the tables. We have added Fig 2 and Fig 3 to depict the changes that occurred for the measures in the two domains between baseline and endline. We have added the captions to these figures on page 18

Discussion and conclusion:

Please split the section into “Discussion” and “Conclusion and Recommendation”.

We appreciate this thoughtful suggestion. We have adopted this recommendation in the revised manuscript.

Will be good to be concise and to ensure discussion is within the scope and findings of this study.

This recommendation has been adopted.

Reviewers' comments:

Reviewer's Responses to Questions

Comments to the Author

1. Is the manuscript technically sound, and do the data support the conclusions?

Reviewer #1: Yes

Reviewer #2: Partly

Shape

2. Has the statistical analysis been performed appropriately and rigorously?

Reviewer #1: Yes

Reviewer #2: Yes

Shape

3. Have the authors made all data underlying the findings in their manuscript fully available?

Reviewer #1: Yes

Reviewer #2: Yes

Shape

4. Is the manuscript presented in an intelligible fashion and written in standard English?

Reviewer #1: Yes

Reviewer #2: Yes

Shape

5. Review Comments to the Author

Reviewer #1: The study addresses an important gap by examining the effects of integrated economic empowerment and sexual and reproductive health (SRH) interventions on adolescent girls in Nigeria. This is particularly valuable given the unique challenges faced by this demographic.

However there are some improvements I would recommend:

Abstract Clarity: The abstract is dense, especially in terms of results. It would be beneficial to streamline the findings to highlight the most impactful outcomes.

We have revised the abstract on page 2 by removing most of the numbers and rewording the results to reduce redundancy.

Introduction - Background Context: The background is comprehensive but could benefit from a more focused framing of the economic and SRH challenges specifically faced by adolescent girls in Nigeria. Additionally, contextualizing this intervention in the broader scope of Sub-Saharan Africa would help readers unfamiliar with the region.

We appreciate this feedback. We have been able to contextualize the background information to the context where the study was conducted and to the sub-Saharan Africa Context on page 3.

Methodological Details:

• Sampling and Recruitment: More details about the sampling process, including any potential biases, would enhance the transparency and replicability of the study

We are grateful for this thoughtful suggestion. We have reorganized the participant recruitment sub-section on page 9 and 10 to improve the flow and provided statements to clarify the sampling procedures. Selection bias could have occurred with the intervention group, because mobilisers might have preferred to select girls who were relatives, daughters of their friends or from the same religious group, ethnicity or clan with a motivation to ensure that the intervention reached those closer to them. We conducted a differential analysis comparing baseline characteristics between participants in the intervention and comparison group and found some differences (Table 2). To mitigate the influence of these differences in the analysis models, we included these variables as covariates in the difference-in-difference models. We have included a statement about selection bias under the study limitations on page 27.

• Covariates and Confounding: The manuscript mentions adjustments for certain covariates. It would be helpful to clarify why specific variables (e.g., age, education) were chosen and to justify their relevance in the analysis.

Under the sub-section on statistical analysis, we have included a statement on page 14 explaining how the covariates were selected. Briefly, evidence such as that from the synthesis by Stavropoulou M., (2018) identified age, parity and marital status as crucial determinants of the success of economic empowerment programs involving adolescent girls. We also considered variables as covariates if there were baseline group differences for this variable between the intervention and comparison group and this has been cited in the statement we have added.

Discussion and Implications:

Interpretation of Findings: The discussion could benefit from more critical reflection on why certain outcomes varied significantly between Kaduna and Ogun, particularly regarding SRH outcomes. Exploring cultural, social, or structural factors could provide readers with a more nuanced understanding.

Under the discussion section, we have included statements to illustrate why for some outcomes, intervention effects were detected in Kaduna or Ogun and not in both states such as line 413-416. We believe that differences in the age groups of the participants and marital status might account for the observed differences. Furthermore, our intention was not to compare outcomes between these groups because they are uniquely different and how the interventions were implemented was also contextualized to their life circumstances.

Program Duration and Sustainability: The study was conducted over a nine-month period, which limits long-term outcome assessment. Discussing the sustainability of these outcomes beyond the study period would be valuable.

We acknowledge that the sustainability of the intervention effects is a challenge with our evaluation. This program was designed to be implemented in a short time to demonstrate the feasibility of implementation and to illustrate potential for showing positive outcomes in the short-term. The program could not be continued for a long term because the investment by the financial partners was not availed. We have cited our inability to show the legacy of these outcomes as a limitation of our study under study limitations on lines 485-489 page 27.

Typographical Errors: A thorough proofreading would help eliminate minor typographical errors and improve readability.

We regret this error. We have rigorously reviewed the draft to increase accuracy and reduced wordiness to increase readability.

Reviewer #2: 

Expand the Timeline

The study observed changes over nine months, which might not fully capture the long-term effects of the intervention. Extending the study period would provide more robust insights into sustained outcomes, particularly in economic empowerment and contraceptive use.

We acknowledge this feedback. The long-term sustainability of the outcomes observed for economic empowerment interventions is a predominant concern for players in this sector. As we have responded to the second reviewer, we have cited the study’s inability to demonstrate the durability of the detected intervention effects as a limitation of our study. Our study was limited to nine months because of lack of financial resources to continue the investment beyond that. We take this as a missed opportunity because it is probable that our intervention contributed to larger and sustained outcomes in the medium and long-term as documented in unpublished data from qualitative interviews conducted with participants, mentors and session facilitators and husbands which affirmed the observed findings through increased involvement income generating activities and transformation of the girl’s decision-making power, agency and self efficacy. This is an important consideration that warrants additional research.

Address Attrition Issues

• Attrition rates were high, especially in Ogun. Future studies could employ strategies such as incentivizing participation, increasing community engagement, or using mobile follow-ups to reduce loss to follow-up and ensure a more representative sample at endline.

We appreciate the suggestions about how we could have mitigated the high attrition rates in Ogun. We explored some of these interventi

---

## [Decision Letter · Decision Letter 1]

1 Aug 2025

Integrated economic and sexual and reproductive health programming among married and unmarried adolescent girls in Nigeria: results from a quasi-experimental cohort study

PONE-D-24-34510R1

Dear Dr. Philps,

We’re pleased to inform you that your manuscript has been judged scientifically suitable for publication and will be formally accepted for publication once it meets all outstanding technical requirements. Please disregard comments from reviewer #3. 

Kind regards,

Jahun Ibrahim, MD, MSC, PhD

Academic Editor

PLOS ONE

Additional Editor Comments (optional):

Reviewers' comments:

Reviewer's Responses to Questions

**Comments to the Author**

Reviewer #3: All comments have been addressed

Reviewer #4: All comments have been addressed

2. Is the manuscript technically sound, and do the data support the conclusions?

Reviewer #3: Partly

Reviewer #4: Yes

3. Has the statistical analysis been performed appropriately and rigorously?

Reviewer #3: Yes

Reviewer #4: Yes

4. Have the authors made all data underlying the findings in their manuscript fully available?

Reviewer #3: Yes

Reviewer #4: Yes

5. Is the manuscript presented in an intelligible fashion and written in standard English?

Reviewer #3: Yes

Reviewer #4: Yes

Reviewer #3: • The figures and tables are informative—ensure consistent labeling and that all are referenced in the text.

• Double-check all acronyms (e.g., SRH, EE) are spelled out on first use in each section.

• Carefully proofread again for minor grammatical inconsistencies (some still present in the revised version).

Final Assessment

thoroughly to reviewer comments.

• Well-structured, methodologically sound, policy-relevant.

• Minor issues:

• Results remain text-heavy.

• Slight inconsistencies in data interpretation across sections (e.g., Ogun attrition vs effect size).

Recommendation:

• Suitable for publication after minor textual refinements and attention to figure/table integration.

Reviewer #4: The title of the paper needs to be fine-tuned. You may change it to 'Integrated Economic and Sexual & Reproductive Health Programming for Married and Unmarried Adolescent Girls in Nigeria: Findings from a Quasi-Experimental Cohort Study'

what does this mean?). If published, this will include your full peer review and any attached files.

**Do you want your identity to be public for this peer review?** For information about this choice, including consent withdrawal, please see our Privacy Policy

Reviewer #3: No

Reviewer #4: **Yes: ** Mukhtar Liman Ahmed

---

## [Editor Report · Acceptance letter]

PONE-D-24-34510R1

PLOS ONE

Dear Dr. Phillips,

I'm pleased to inform you that your manuscript has been deemed suitable for publication in PLOS ONE. Congratulations! Your manuscript is now being handed over to our production team.

Kind regards,

on behalf of

Dr. Ibrahim Jahun

Academic Editor

PLOS ONE